# Dynamic Weight Strategy of Physics-Informed Neural Networks for the 2D Navier–Stokes Equations

**DOI:** 10.3390/e24091254

**Published:** 2022-09-06

**Authors:** Shirong Li, Xinlong Feng

**Affiliations:** College of Mathematics and System Sciences, Xinjiang University, Urumqi 830046, China

**Keywords:** physics-informed neural networks, dynamic weight strategy, Navier–Stokes equations

## Abstract

When PINNs solve the Navier–Stokes equations, the loss function has a gradient imbalance problem during training. It is one of the reasons why the efficiency of PINNs is limited. This paper proposes a novel method of adaptively adjusting the weights of loss terms, which can balance the gradients of each loss term during training. The weight is updated by the idea of the minmax algorithm. The neural network identifies which types of training data are harder to train and forces it to focus on those data before training the next step. Specifically, it adjusts the weight of the data that are difficult to train to maximize the objective function. On this basis, one can adjust the network parameters to minimize the objective function and do this alternately until the objective function converges. We demonstrate that the dynamic weights are monotonically non-decreasing and convergent during training. This method can not only accelerate the convergence of the loss, but also reduce the generalization error, and the computational efficiency outperformed other state-of-the-art PINNs algorithms. The validity of the method is verified by solving the forward and inverse problems of the Navier–Stokes equation.

## 1. Introduction

Numerical simulation of fluid systems relies on solving partial differential equations using computational fluid dynamics (CFD) methods, including finite element, finite volume and finite difference methods [1,2,3]. However, it is usually expensive and time-consuming to generate meshes for solving equations in complex regions. Solving the inverse problem by CFD methods first requires tedious data assimilation and does not guarantee convergence [4]. Therefore, the use of CFD models in practical applications and real-time predictions is limited. It is important to develop an effective Navier–Stokes solver that can overcome these limitations.

In recent years, deep neural networks have received extensive attention in the field of scientific machine learning [5]. It can be used to construct new methods for solving partial differential equations, based on their well-known capability as universal function approximators [6]. The process of analyzing complex physical and engineering systems usually requires a large amount of data. The cost of data acquisition is often prohibitive, and we inevitably face the challenge of making decisions with partial information. When the governing equation is known, a neural network trained under the constraints of this equation can learn the solution of the partial differential equation with a small amount of data. The potential of using neural networks to solve partial differential equations has been recognized. With huge advances in computational power and training algorithms [7], and the invention of automatic differentiation methods [8], physics-informed neural networks(PINNs) were able to take this approach to a different level [9]. PINNs does not require mesh generation to solve fluid mechanics problems [10,11,12,13], so it has advantages for solving problems in complex regions. This is a good strategy to tackle the problem of the curse of dimensionality.

The baseline PINNs has provided new research ideas for solving Navier–Stokes equations, stochastic PDEs, and fractional PDEs [14,15,16,17]. However, it has been argued that the convergence and accuracy of PINNs are still of tremendous challenge, especially for Navier–Stokes with multi-scale characteristics. Wang [18] explored the problems existing in the training process of PINNs, and found that the gradient value of each residual term in the loss was greatly different in the process of backpropagation, and the training could not be balanced. Therefore, a method of adaptively adjusting the weights between different components in the loss is proposed to improve the training convergence. The problems of gradient vanishing and gradient explosion limit the application of this method. Zhao et al. also observed that the performance of PINNs is closely related to the appropriate combination of loss terms [10]. It introduces a non-adaptive weighting strategy and time adaptive strategy of loss function. However, using fixed weights is always time-consuming, labor-intensive, and prone to errors and omissions. If gradient descent optimizes multiple objectives consisting of fixed weights, there is a high probability of obtaining a local optimal solution [19]. Ref. [20] introduced adaptive weights for configuration and boundary losses, which are updated through the Neural Tangent Kernel (NTK). The distribution of the eigenvalues of NTK does not change, and it is more difficult to calculate the eigenvalues of NTK. The performance improvement is slight. A method that updates the adaptation weights of configuration points in the loss function concerning the network parameters was suggested [21]. It performs well for the AC equation. For other equations, the objective function has a convergence problem due to the existence of the Max system. A strategy for adaptive resampling of configuration points is proposed to improve the convergence and accuracy of some partial differential equations with steep gradient solutions [22]. After each resampling step, the number of residual points grows, increasing computational complexity.

This paper introduces a dynamic weight strategy for physics-informed neural networks(dwPINNs) to balance the contribution of each loss item to the network. The mechanism of weight update in this paper is completely different from other PINNs literature. It uses trainable weights as a soft multiplicative mask reminiscent of the attention mechanism used in computer vision [23]. Adaptive weights are trained simultaneously with network parameters, and the data are automatically weighted in the loss function, forcing the approximation of these data to improve. This is achieved by training the network to minimize loss and maximize weights. The weight update is based on the information of mean square error. This strategy is conducive to the convergence of loss and can reduce the generalization error.

This paper is organized as follows: Section 1 is the Introduction, and Section 2 gives a brief discussion of the effect of the loss terms on the network parameters in the optimization method. In Section 3 we discuss the construction of the loss function and some theoretical results on the convergence of weight sequence in the proposed method. Section 4 gives the results and detailed discussions on dwPINNs solving the Navier–Stokes. Finally, in Section 5, we summarize the conclusions of our work.

## 2. Preliminaries

### 2.1. Partial Differential Equations

The incompressible Navier–Stokes equations is:
(1a)$$\begin{array}{cc} \frac{\partial u}{\partial t}+\lambda_{1}\left(u\cdot \nabla \right)u+\nabla p-\lambda_{2}\nabla^{2}u & =f\;\;\;\;\mathrm{in}\;\Omega \times [0,T], \end{array}$$
(1b)$$\begin{array}{cc} \;\nabla \cdot u & =0\;\;\;\;\mathrm{in}\;\Omega \times [0,T], \end{array}$$
(1c)$$\begin{array}{cc} \;u(x,0) & =u_{0}\;\;\mathrm{in}\;\Omega , \end{array}$$
(1d)$$\begin{array}{cc} \;u & =u_{b}\;\;\mathrm{on}\;\Gamma_{D}\times \left[0,T\right]. \end{array}$$
where u(t,x)=(u(t,x),v(t,x)) is a velocity vector field, *p* is a scalar pressure field. $\lambda_{1},\lambda_{2}$ are the unknown parameters, and Ω is the solution domain. $\Gamma_{D}$ is the boundaries of the computational domain. Formula ([Disp-formula FD1a-entropy-24-01254]) is the Momentum equation, Formula ([Disp-formula FD1a-entropy-24-01254]) is the conservation of mass equation, Formula ([Disp-formula FD1a-entropy-24-01254]) is Initial conditions, and Formula ([Disp-formula FD1a-entropy-24-01254]) is Dirichlet boundary conditions. In this paper, we shall solve both forward problems where solutions of partial differential equations are inferred given fixed model parameters $\lambda_{1},\lambda_{2}$ as well as inverse problems, where the unknown parameters $\lambda_{1},\lambda_{2}$ are learned from the observed data.

### 2.2. Fully Connected Neural Networks

Neural networks have well-known function approximation properties [24], so they can be used to approximate the solution of partial differential equations. The (L−1) -hidden layer feed-forward neural network is defined by
$$Z^{k}\left(x\right)=W^{k}\Phi \left(Z^{k-1}\left(x\right)\right)+b^{k}\in R^{N_{k}},\;2\leq k\leq L,$$
and $Z^{1}\left(x\right)=W^{1}x+b^{1},$ where in the last layer, the activation function is identity. By letting $\tilde{\Theta }=\left(W^{k},b^{k}\right)$ as the collection of all weights, biases. Φ is a nonlinear activation function. We can write the output of the neural network as
$$u_{\tilde{\Theta }}\left(x\right)=Z^{L}\left(x;\tilde{\Theta }\right),$$
where $Z^{L}\left(x;\tilde{\Theta }\right)$ emphasizes the dependence of the neural network output $Z^{L}\left(x\right)$ on $\tilde{\Theta }$. The loss function can be defined as
$$L=\min_{\tilde{\Theta }}\frac{1}{N_{u}}\sum_{i=1}^{N_{u}}{\left|u\left(x_{u}^{i},\tilde{\Theta }\right)-u^{i}\right|}^{2}.$$

Here, ${\left\{x_{u}^{i},u^{i}\right\}}_{i=1}^{N_{u}}$ denote the training data on u(x). $N_{u}$ is the number of training data. The training process requires the loss function minimized with respect to the weights and biases in each network layer. Fitting in the sense of least squares requires the minimum mean square error between the fitting function and original data, and it is an approximation in the overall sense.

### 2.3. Optimization Method

We seek to find $\tilde{\Theta^{*}}$ that minimizes the loss function $L\left(\tilde{\Theta }\right)$. There are several optimization algorithms available to minimize the loss function. In general, a gradient-based optimization method is employed for the training of parameters [25]. Weights and biases are initialized from known probability distributions. In the basic form, given an initial value of parameters $\tilde{\Theta }$ using Xavier initialization, the parameters are updated as
$${\tilde{\Theta }}^{m+1}={\tilde{\Theta }}^{m}-{\left.\eta_{l}\frac{\partial L\left(\tilde{\Theta }\right)}{\partial \tilde{\Theta }}\right|}_{\tilde{\Theta }={\tilde{\Theta }}^{m}},$$
where $\eta_{l}$ is the learning rate. ${\tilde{\Theta }}^{m}$ is the network parameter for step *m*.

Next, we illustrate the effect of the error on the gradient in backpropagation. The gradient of the loss function to the network parameters in the backpropagation algorithm [26] is as follows
(2a)$$\begin{array}{cc} \; & \delta^{\left(L\right)}=-\left(u-a^{\left(L\right)}\right)⊙f^{\prime }\left(z^{\left(L\right)}\right), \end{array}$$
(2b)$$\begin{array}{cc} \; & \delta^{\left(l\right)}=\left({\left(W^{\left(l+1\right)}\right)}^{⊤}\delta^{\left(l+1\right)}\right)⊙f^{\prime }\left(z^{\left(l\right)}\right), \end{array}$$
(2c)$$\begin{array}{cc} \; & \frac{\partial L}{\partial W^{\left(l\right)}}=\delta^{\left(l\right)}{\left(a^{\left(l-1\right)}\right)}^{⊤}, \end{array}$$
(2d)$$\begin{array}{cc} \; & \frac{\partial L}{\partial b^{\left(l\right)}}=\delta^{l}. \end{array}$$

⊙ represents the Hadamard product; $n_{l}$ represents the number of neurons in layer *l*; f(·) represents the activation function of the neuron; $W^{\left(l\right)}\in R^{n_{l}\times n_{l-1}}$ represents the weight matrix from l−1 to *l*; $b^{\left(l\right)}={\left(b_{1}^{\left(l\right)},b_{2}^{\left(l\right)},\cdots ,b_{n_{l}}^{\left(l\right)}\right)}^{⊤}\in R^{n_{l}}$ represents the bias from l−1 layer to *l* layer; $z^{\left(l\right)}={\left(z_{1}^{\left(l\right)},z_{2}^{\left(l\right)},z_{n_{l}}^{\left(l\right)}\right)}^{⊤}\in R^{n_{l}}$ represents the output of neurons in the *l* layer; $a^{\left(l\right)}={\left(a_{1}^{\left(l\right)},a_{2}^{\left(l\right)},\cdots ,a_{n_{l}}^{\left(l\right)}\right)}^{⊤}\in R^{n_{l}}$ represents the activation value of neurons in the *l* layer. Formula ([Disp-formula FD1a-entropy-24-01254]) is the gradient of the loss function to the weight of the output layer and the hidden layer. Formula ([Disp-formula FD1a-entropy-24-01254]) is the gradient of the loss function to the bias of the output layer and the hidden layer. We calculate the $\delta^{\left(l\right)}$ of the *l* layer through the $\delta^{\left(l+1\right)}$ of the l+1 layer, and then obtain the gradient of the weight of the hidden layer by the Formula ([Disp-formula FD1a-entropy-24-01254]).

The error has a great influence on the backpropagation gradient, which can be obtained from ([Disp-formula FD1a-entropy-24-01254]). When the objective function consists of many terms, it always tends to optimize the loss term with a larger error. Formulas ([Disp-formula FD1a-entropy-24-01254]) and ([Disp-formula FD1a-entropy-24-01254]) shows that the more hidden layers in the network, the easier the problem of gradient disappearance and explosion. Therefore, the network structure used in the experiment is a wide and shallow structure, and the number of hidden layers does not exceed five layers.

## 3. Methodology

The combination of multiple loss functions plays a significant role in the convergence of PINNs [10]. The most common way to combine losses of each constraint is the weighted summation. These are either nonadaptive or require training many times at an increased computational cost. Here, we propose a simple procedure using fully trainable weights. It is in line with the idea of neural network adaptation, that is, the dynamic weights in the loss function are updated together with network parameters through backpropagation.

### 3.1. Dynamic Weights Strategy for Physics-Informed Neural Networks

We define residuals to be given by the left-hand-side of Equations ([Disp-formula FD1a-entropy-24-01254]) and ([Disp-formula FD1a-entropy-24-01254]); i.e.,
(3)$$f_{1}:=\frac{\partial u}{\partial t}+\lambda_{1}\left(u\cdot \nabla \right)u+\nabla p-\lambda_{2}\nabla^{2}u-f,f_{2}:=\nabla \cdot u.$$
and proceed by approximating u(t,x) by neural networks. This assumption, along with Equation ([Disp-formula FD1a-entropy-24-01254]) results in physical constraints $f_{1},f_{2}$. $f_{1}$ indicates that the numerical solution satisfies the conservation of momentum, and $f_{2}$ indicates that the numerical solution satisfies the conservation of mass. The physical constraints of the network can be derived by applying the chain rule for differentiating compositions of functions using automatic differentiation. In order to balance the training of the residuals in each part of the loss, we multiply the trainable weights before each residual term of the PINNs loss function. The objective function is defined as follows
(4)$$J=w_{u}MSE_{u}+w_{0}MSE_{0}+w_{b}MSE_{b}+w_{f}MSE_{f},$$
where
$$MSE_{u}=\frac{1}{N_{u}}\sum_{i=1}^{N_{u}}{\left|u\left(t_{u}^{i},x_{u}^{i},\tilde{\Theta }\right)-u_{u}^{i}\right|}^{2},$$
$$MSE_{0}=\frac{1}{N_{0}}\sum_{i=1}^{N_{0}}{\left|u\left(t_{0}^{i},x_{0}^{i},\tilde{\Theta }\right)-u_{0}^{i}\right|}^{2},$$
$$MSE_{b}=\frac{1}{N_{b}}\sum_{i=1}^{N_{b}}{\left|u\left(t_{b}^{i},x_{b}^{i},\tilde{\Theta }\right)-u_{b}^{i}\right|}^{2},$$
and
$$MSE_{f}=\frac{1}{N_{f}}\sum_{i=1}^{N_{f}}\left({\left|f_{1}\left(t_{f}^{i},x_{f}^{i},\tilde{\Theta }\right)\right|}^{2}+{\left|f_{2}\left(t_{f}^{i},x_{f}^{i},\tilde{\Theta }\right)\right|}^{2}\right).$$

Here, $w_{u},w_{0},w_{b},w_{f}$ are the newly introduced balance parameters. ${\left\{t_{u}^{i},x_{u}^{i},u^{i}\right\}}_{i=1}^{N_{u}}$ denote observed data (if any) ${\left\{t_{0}^{i},x_{0}^{i},u_{0}^{i}\right\}}_{i=1}^{N_{0}},{\left\{t_{b}^{i},x_{b}^{i},u_{b}^{i}\right\}}_{i=1}^{N_{b}}$ denote the initial and boundary training data on u(t,x) and ${\left\{t_{f}^{i},x_{f}^{i}\right\}}_{i=1}^{N_{f}}$ specify the collocations points for $f_{1}\left(t,x\right),f_{2}\left(t,x\right)$. $N_{u},N_{0},N_{b},N_{f}$ is the number of corresponding data. The sampling method of initial and boundary training data is random selection at the corresponding boundary. The selection method of the collocations points is Latin hypercube sampling. We determine unknown parameters by performing the following tasks
(5)$$\min_{\tilde{\Theta }}\max_{w_{u},w_{0},w_{b},w_{f}}J\left(\tilde{\Theta },w_{u},w_{0},w_{b},w_{f}\right).$$

This can be accomplished by a gradient descent/ascent procedure, with updates given by
(6a)$$\begin{array}{cc} {\tilde{\Theta }}^{k+1} & ={\tilde{\Theta }}^{k}-\eta_{k}\nabla_{\tilde{\Theta }}J\left({\tilde{\Theta }}^{k},w_{u}^{k},w_{f}^{k},w_{b}^{k},w_{0}^{k}\right), \end{array}$$
(6b)$$\begin{array}{cc} w_{u}^{k+1} & =w_{u}^{k}+\eta_{w}^{k}\nabla_{w_{u}}J\left({\tilde{\Theta }}^{k},w_{u}^{k},w_{f}^{k},w_{b}^{k},w_{0}^{k}\right), \end{array}$$
(6c)$$\begin{array}{cc} w_{0}^{k+1} & =w_{0}^{k}+\eta_{w}^{k}\nabla_{w_{0}}J\left({\tilde{\Theta }}^{k},w_{u}^{k},w_{f}^{k},w_{b}^{k},w_{0}^{k}\right), \end{array}$$
(6d)$$\begin{array}{cc} w_{b}^{k+1} & =w_{b}^{k}+\eta_{w}^{k}\nabla_{w_{b}}J\left({\tilde{\Theta }}^{k},w_{u}^{k},w_{f}^{k},w_{b}^{k},w_{0}^{k}\right), \end{array}$$
(6e)$$\begin{array}{cc} w_{f}^{k+1} & =w_{f}^{k}+\eta_{w}^{k}\nabla_{w_{f}}J\left({\tilde{\Theta }}^{k},w_{u}^{k},w_{f}^{k},w_{b}^{k},w_{0}^{k}\right), \end{array}$$
where $\eta_{k}$ is the learning rate for the *k*th step in the process of updating the network parameters, $\eta_{w}^{k}$ is the learning rate for the *k*th step in the process of updating the balance parameters. Considering the dynamic weight $w_{0}$, to fix ideas, we see that
(7)$$\nabla_{w_{0}}J\left({\tilde{\Theta }}^{k},w_{u}^{k},w_{0}^{k},w_{b}^{k},w_{f}^{k}\right)=MSE_{0}^{k}\geq 0,$$

The sequence of weights $\left\{w_{0}^{k};k=1,2\ldots \right\}$ is monotonically nondecreasing, provided that $w_{0}^{1}$ is initialized to a non-negative value. Furthermore, (Equation 7) shows that the magnitude of the gradient, and therefore of the update, is larger when the mean squared error $MSE_{0}^{k}$ is large. This progressively penalizes the network more for not fitting the initial points closely. Notice that any of the weights can be set to fixed, non-trainable values if desired. For example, by setting $w_{b}^{k}\equiv 1$, only the weights of the initial and collocation points would be trained. If necessary, the weight can also be changed to other types of functions. The convergence of the weight sequence plays an important role in the stability of the Min system. Next, we prove the convergence of $w_{0}^{k}$. From ([Disp-formula FD1a-entropy-24-01254]), we can obtain
$$\left|w_{0}^{k+1}-w_{0}^{k}\right|=\eta_{w}^{k}\nabla_{w_{0}}J\left({\tilde{\Theta }}^{k},w_{u}^{k},w_{f}^{k},w_{b}^{k},w_{0}^{k}\right),$$
$$\left|w_{0}^{k}-w_{0}^{k-1}\right|=\eta_{w}^{k-1}\nabla_{w_{0}}J\left({\tilde{\Theta }}^{k-1},w_{u}^{k-1},w_{f}^{k-1},w_{b}^{k-1},w_{0}^{k-1}\right).$$
According to (Equation 7)
$$\left|w_{0}^{k+1}-w_{0}^{k}\right|=\eta_{w}^{k}MSE_{0}^{k},\left|w_{0}^{k}-w_{0}^{k-1}\right|=\eta_{w}^{k-1}MSE_{0}^{k-1},$$
(8)$$\eta_{w}^{k}MSE_{0}^{k}\leq \eta_{w}^{k-1}MSE_{0}^{k-1}.$$

Therefore $\left|w_{0}^{k+1}-w_{0}^{k}\right|\leq \left|w_{0}^{k}-w_{0}^{k-1}\right|$. According to the principle of compression mapping, $\left\{w_{0}^{k}\right\}$ is convergent. $\left\{w_{0}^{k}\right\}$ has an upper bound. The analysis for $w_{u}^{k}$,$w_{b}^{k}$,$w_{f}^{k}$ is the same as $w_{0}^{k}$.

**Remark** **1.**
*From Formula (Equation 8), it can be seen that the convergence of the sequence $\left\{w_{0}^{k}\right\}$ depends on the monotonous decrease in the $MSE_{0}^{k}$. The mean square error is theoretically monotonically decreased, so the weight sequence is theoretically convergent. However, in actual training, the mean square error is not strictly monotonically decreasing, which may have a little adverse effect on the performance of our method.*


In order to overcome the above problems as much as possible, Algorithm is adopted. To strengthen the condition of balancing weight update, we update the weight only when the condition $MSE_{0}^{k+1}<MSE_{0}^{k}$ is satisfied. As far as possible, to make the weight sequence meet the convergence conditions, reducing the fluctuation of MSE has an adverse effect on our method. In our implementation of dwPINNs, we use Tensorflow with a fixed number of iterations of Adam followed by another fixed number of iterations of the L-BFGS quasi-newton method [27,28]. This is consistent with the PINNs formulation in [9], as well as follow-up literature [11]. The adaptive weights are only updated in the Adam training steps and are held constant during L-BFGS training. The dwPINNs algorithm is summarized.
**Algorithm 1: **Dynamic weights strategy for PINNs$$\begin{array}{c} \;\mathrm{Step}\;1:\;\mathrm{Set}\;\mathrm{Training}\;\mathrm{steps}\;K,\;\mathrm{the}\;\mathrm{learning}\;\mathrm{rate}\;\eta ,\eta_{w},\;\mathrm{initial}\;\mathrm{values}\;\mathrm{balance}\;\mathrm{weights}\;\\ w=\left\{w_{u},w_{0},w_{b},w_{f}\right\}\;\mathrm{and}\;\mathrm{neural}\;\mathrm{network}\;\mathrm{parameters}\;\tilde{\Theta }.\\ \;\mathrm{Step}\;2:\;\mathrm{Consider}\;a\;\mathrm{physics}-\mathrm{informed}\;\mathrm{neural}\;\mathrm{network}\;\;\mathrm{to}\;\mathrm{define}\;\mathrm{the}\;\mathrm{weighted}\;\mathrm{loss}\;\mathrm{function}\;\\ J\left(\tilde{\Theta },w_{u},w_{0},w_{b},w_{f}\right)\;\mathrm{based}\;\mathrm{on}\;\left(\right).\;\\ \;\mathrm{Step}\;3:\;\mathrm{Then}\;\mathrm{use}\;K\;\mathrm{steps}\;\mathrm{of}\;a\;\mathrm{gradient}\;\mathrm{descent}\;\mathrm{algorithm}\;\;\mathrm{to}\;\mathrm{update}\;\mathrm{the}\;\mathrm{parameters}\;w\\ \;\mathrm{and}\;\tilde{\Theta }\;\mathrm{as}:\;\\ \;\mathrm{for}\;k=1\;\mathrm{to}\;K\;\mathrm{do}\;\\ \;\;\mathrm{if}\;MSE_{0}^{k+1}<MSE_{0}^{k}\;\mathrm{and}\;MSE_{u}^{k+1}<MSE_{u}^{k}\;\mathrm{and}\;MSE_{b}^{k+1}<MSE_{b}^{k}\;\mathrm{and}\;MSE_{f}^{k+1}<MSE_{f}^{k}\\ \;\;\;\mathrm{Tune}\;\mathrm{the}\;\mathrm{balance}\;\mathrm{weights}\;w\;\mathrm{via}\;\mathrm{Adam}\;\mathrm{to}\;\mathrm{maximize}\;\mathrm{the}\;\mathrm{meeting}\;\mathrm{constraints}\\ \;\;\;\;\;\;\;\;\;w_{k+1}\leftarrow \;\mathrm{Adam}1\;\left(J\left(w_{k};\tilde{\Theta };K\right);\eta ;\eta_{w}\right)\\ \;\;\mathrm{Update}\;\mathrm{the}\;\mathrm{parameters}\;\tilde{\Theta }\;\mathrm{via}\;\mathrm{Adam}\;\mathrm{to}\;\mathrm{minimize}\;J\\ \;\;\;\;\;\;\;\;\;{\tilde{\Theta }}_{k+1}\leftarrow \;\mathrm{Adam}2\;\left(J\left(w_{k};{\tilde{\Theta }}_{k};K\right);\eta ;\eta_{w}\right)\\ \;\mathrm{end}\;\mathrm{for}\; \end{array}$$

### 3.2. A Brief Note on the Errors Involved in the dwPINNs Methodology

Let F and *u* be the family of functions that can be represented by the chosen neural network and the exact solution of PDE. Then, we define $u_{a}=\arg\min_{f\in F}∥f-u∥$ as the best approximation to the exact solution *u*. Let $u_{g}=\arg\min_{\tilde{\Theta }}J\left(\tilde{\Theta }\right)$ be the solution of net at global minimum and $u_{t}=\arg\min_{\tilde{\Theta }}J\left(\tilde{\Theta }\right)$ be the solution of net at local minimum. Therefore, the total error consists of an approximation error $E_{\mathrm{app}\;}=∥u-u_{a}∥,$ the optimization error $E_{\mathrm{opt}\;}=∥u_{t}-u_{g}∥$ and the generalization error $E_{\mathrm{gen}\;}=∥u_{g}-u_{a}∥.$ In PINNs, the number and location (distribution) of residual points are two important factors that affect the generalization error [13]. The optimization error is introduced due to the complexity of the loss function. The performance of PINNs is closely related to the appropriate combination of loss terms, which may avoid local optimization. Thus, the total error in PINNs as
(9)$$E_{PINN}:=∥u_{t}-u∥\leq ∥u_{t}-u_{g}∥+∥u_{g}-u_{a}∥+∥u_{a}-u∥.$$

### 3.3. Advantages of Dynamic Weight Strategy for Physics-Informed Neural Networks

1.The optimization error can be reduced by using the dynamic weight strategy for physics-informed neural networks. During training, each part of the loss function can be dropped more evenly, and the loss can become smaller and converge faster.2.This method can reduce the generalization error by increasing the weights of hard-to-train points during training. It also makes the error of such hard-to-train points smaller.

## 4. Numerical Examples

In this section We apply the proposed dwPINNs to simulate different incompressible Navier–Stokes flows. First, we consider two-dimensional unsteady equations with the analytic solution to investigate the effectiveness of dwPINNs. Then, we employ the dynamic weights strategy to steady lid-driven cavity flow in two dimensions. Compared with other PINNs methods, the efficiency of this method is highlighted. Finally, the inverse problem of the flow around a cylinder is solved by the dwPINNs method. In the numerical experiments, we strictly control the non-experimental variables of the two test methods to be the same, such as training data, network structure, optimization method, etc. The activation functions used in the following numerical experiments are tanh. G.Cybenkot [29] found that it has a strong linear superposition approximation ability and is more suitable for function approximation. We keep the amount of training data roughly equal to the parameters of the neural network to avoid the overfitting problem. To illustrate the efficiency of the proposed method, the accuracy of the trained model is assessed through the relative $L_{2}$ error of the exact value $\bar{u}\left(x_{i},t_{i}\right)$ and the trained approximation $u(x_{i},t_{i})$ inferred by the network at the data ${\left\{t^{i},x^{i}\right\}}_{i=1}^{N_{t}}$, $N_{t}$ is the number of test data. The deep learning framework used in this experiment is tensorflow2.3. In terms of hardware, the CPU is Intel CORE i5 7th Gen, the memory is 4G, and the GPU is NVIDIA GeForce 940MX.

### 4.1. Navier–Stokes Equations with Analytic Solution

We first solve forward problems. We use the 2D unsteady Navier–Stokes with an analytical solution as the first test case to demonstrate the feasibility of dwPINNs. The analytical solution is as follows
(10)$$\begin{array}{cc} u^{*} & =-\sin\left(t\right)\sin^{2}\left(\pi x\right)\sin\left(\pi y\right)\cos\left(\pi y\right),\\ v^{*} & =\sin\left(t\right)\sin\left(\pi x\right)\cos\left(\pi x\right)\sin^{2}\left(\pi y\right),\\ p^{*} & =\sin(t)\sin(\pi x)\cos(\pi y). \end{array}$$

Based on Section 3.1, the problem loss is defined as follows
(11)$$\begin{array}{c} {MSE}_{0}=\frac{1}{N_{0}}\sum_{i=1}^{N_{0}}{\left|u\left(t_{0}^{i},x_{0}^{i},\tilde{\Theta }\right)-u_{0}^{i}\right|}^{2},\\ {MSE}_{b}=\frac{1}{N_{b}}\sum_{i=1}^{N_{b}}{\left|u\left(t_{b}^{i},x_{b}^{i},\tilde{\Theta }\right)-u_{b}^{i}\right|}^{2},\\ {MSE}_{f}=\frac{1}{N_{f}}\sum_{i=1}^{N_{f}}\left({\left|f_{1}\left(t_{f}^{i},x_{f}^{i},\tilde{\Theta }\right)\right|}^{2}+{\left|f_{2}\left(t_{f}^{i},x_{f}^{i},\tilde{\Theta }\right)\right|}^{2}\right). \end{array}$$

$\lambda_{1}=1,\lambda_{2}=0.01$. The computational domain is defined by Ω=[0,1]×[0,1] and the time interval is [0,1]. There are 40 points with fixed spatial coordinate on each boundary, $N_{b}=4\times 40$. For computing the equation loss of dwPINNs, 10,000 points are randomly selected inside the domain. Adam training time is 10,000.

The numerical results of the two methods are shown in Table 1. We see that the dwPINNs perform better than the PINNs, and also that applying the dynamic weights can improve the simulation accuracy. A snapshot of the velocity fields together with the absolute errors at t=1 is displayed in Figure 1. It shows that this method is feasible and qualitatively accurate. The convergence of dynamic weights during the training process is displayed in Figure 2.

Next, considering whether dwPINNs can still work in the middle and later stages of training, we design the following experiment. After PINNs are normally trained 5000 times, we conduct dynamic weight strategy training 5000 times and take PINNs as the control group of the experiment. The results are shown in Figure 3. It can be clearly seen from the figure that the dynamic weight strategy can accelerate the convergence of error in the middle and later stages of training.

### 4.2. Comparison of the Different PINNs Methods for 2D Navier–Stokes Equations

The lid-driven cavity flow is a standard test case for verifying the accuracy of new computational methods for incompressible Navier–Stokes equations. Although, there are many papers in the literature that present results of the lid-driven cavity with different formulations, grids and numerical methods, we shall compare the results with Wang [18], where they used a Learning rate annealing for PINNs. The domain is [0,1]×[0,1] and no-slip boundary conditions are applied at the left, bottom and right boundaries. The top boundary moves with constant velocity in the positive *x* direction. u(x)=(u(x),v(x)) is a velocity vector field, *p* is a scalar pressure field. $\lambda_{1}=1,\lambda_{2}=0.01$.

Parameters during training are as follows: $N_{f}=$ 10,000, $N_{b}=3000$. The network structure is a 4-layer deep fully-connected network with 50 neurons per layer. The results of this experiment are summarized in Table 2. It represents average relative L2 error of $\left|u\left(x\right)\right|=\sqrt{u^{2}\left(x\right)+v^{2}\left(x\right)}$ across 10 runs with random restarts. Figure 4 summarizes the results of our experiment, which shows the magnitude of the predicted solution *u*, *v* and absolute point-wise error predicted by dwPINNs, after 32,000 stochastic gradient descent updates using the Adam optimizer. Figure 5 is the convergence process of dynamic weights and the relative L2 error of *u* and *v*.

Learning rate annealing for PINNs is one of the important loss balancing methods [18]. Compared with Learning rate annealing for PINNs, the error of dwPINNs is reduced. The SAPINNs outperformed other state-of-the-art PINN algorithms in L2 error by a wide margin [21]. However, the SAPINNs failed to solve the Navier–Stokes equations. Therefore, the performance of dwPINNs is the best at present.

In order to further analyze the performance of our method, we carried out the following systematic research to quantify its prediction accuracy for different numbers of training points and configuration points, as well as different neural network structures. In Table 3, we report the relative L2 error obtained under the conditions of different initial and boundary training data $N_{u}$ and different configuration points $N_{F}$, while keeping the 4-layer network architecture fixed. With a sufficient number of configuration points $N_{f}$, the overall trend of prediction accuracy continues to improve with the increase in the total number of training data $N_{u}$. Finally, Table 4 shows the resulting relative L2 for different numbers of hidden layers, and different numbers of neurons per layer, while the total number of training and collocation points is kept fixed to $N_{u}=3000$ and $N_{f}=10,000$, respectively. As expected, we observed that the prediction accuracy improved with the increase in the number of layers and neurons.

### 4.3. Inverse Problem: Two-Dimensional Navier-Stokes Equations

Here, we use a dynamic weight strategy for Physics-informed neural networks to simulate the 2D vortex shedding behind a circular cylinder at Re = 100. The cylinder diameter D is 1. The simulation domain size is [0,8]×[−2,−2]. In this example, $\lambda_{1}=1.0,\lambda_{2}=0.01$. High-fidelity data from [9] is used as a reference and for providing training data for dwPINNs. We sample $N_{f}=5000$ collocation points, $N_{u}=5000$ exact points. Our goal is to identify the unknown parameters $\lambda_{1},\lambda_{2}$, as well as to obtain a qualitatively accurate reconstruction of the entire pressure field p(t,x,y) in the cylinder wake, which by definition can only be identified up to a constant. Each loss item is as follows
(12)$$\begin{array}{c} {MSE}_{u}=\frac{1}{N_{u}}\sum_{i=1}^{N_{u}}{\left|u\left(t_{u}^{i},x_{u}^{i},\tilde{\Theta }\right)-u_{u}^{i}\right|}^{2},\\ {MSE}_{f}=\frac{1}{N_{f}}\sum_{i=1}^{N_{f}}\left({\left|f_{1}\left(t_{f}^{i},x_{f}^{i},\tilde{\Theta }\right)\right|}^{2}+{\left|f_{2}\left(t_{f}^{i},x_{f}^{i},\tilde{\Theta }\right)\right|}^{2}\right). \end{array}$$

Numerical results of the Navier–Stokes equation in the way of the dwPINNs are displayed in Table 5. Compared with PINNs, dwPINNs have less training time and higher prediction accuracy. Then, after applying noise to the original training data, the calculation was performed using the dwPINNs method. We observed that even if the training data are corrupted with 1% uncorrelated Gaussian noise, the method can identify unknown parameters very accurately $\lambda_{1}$ and $\lambda_{1}$, indicating that our method has good stability.

Figure 6 shows the loss history of two methods and dynamic weights history. Additionally, the convergence of loss dwPINNs is quicker than PINNs in Figure 6. The scalar $w_{u}$ increases rapidly and is more punitive to the $MSE_{u}$, which leads to faster convergence. The loss of PINNs would attain $9.268\times 10^{-1}$, while dwPINNs would converge to $1.206\times 10^{-1}$. Also plotted are representative snapshots of the predicted velocity components u(t,x,y), v(t,x,y) after the model was trained in Figure 7. Based on the predicted versus instantaneous pressure field p(x,y,t) shown in Figure 7, the error between the predicted value and the true value is extremely low in the entire calculation domain. An interesting result is that, in the absence of any training data on the pressure itself, the network can provide a qualitatively accurate prediction of the entire pressure field p(t,x,y). The neural network architecture used here consists of 4 hidden layers with 50 neurons in each layer. The predicted $\lambda_{1}=1.00065,\lambda_{2}=0.00991$.

## 5. Conclusions

In this paper, we introduced a dynamic weights strategy for loss balanced in physics-informed neural networks. The strategy is helpful to the training of PINNs in solving incompressible Navier–Stokes equations. It can effectively improve unbalanced back-propagated gradients during model training. The most appropriate weights are added to the optimization system through the objective function minimax alternating optimization, which makes the training more balanced. It is theoretically analyzed that the balanced weights are convergent under the condition of monotonic mean square error. The advantages of the method are verified by studying the forward and inverse problems of the Navier–Stokes equation. Various experimental results can support our view that the loss function decays slightly faster and the relative error is lower than the state-of-the-art PINNs. This paper also provides a reference for the application of adaptive mechanisms in the PINNs framework. Code and data accompanying this manuscript are publicly available at https://github.com/1shirong/dwPINNs.git (accessed on 7 July 2022). In this method, the convergence of the maximum system and the convergence of the minimum system are interdependent. Adam and other algorithms can not guarantee to find the global minimum, which will limit the optimal performance of this method. How to design an optimization algorithm specifically for PINNs is an open and meaningful problem.

## Figures and Tables

**Figure 1 entropy-24-01254-f001:**
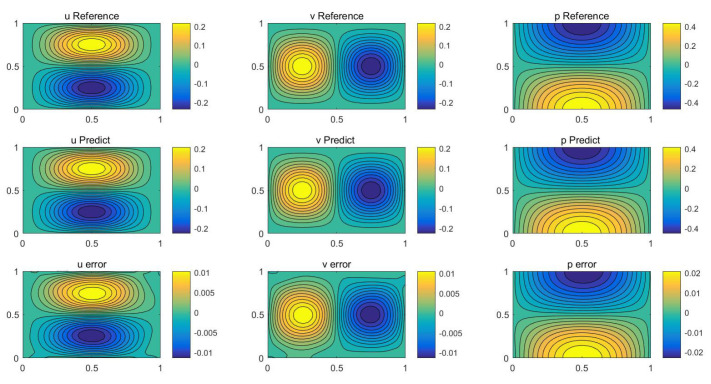
Navier–Stokes: A snapshot of analytical solution (**top**) prediction solution (**middle**) and error (**bottom**) at t=1.

**Figure 2 entropy-24-01254-f002:**
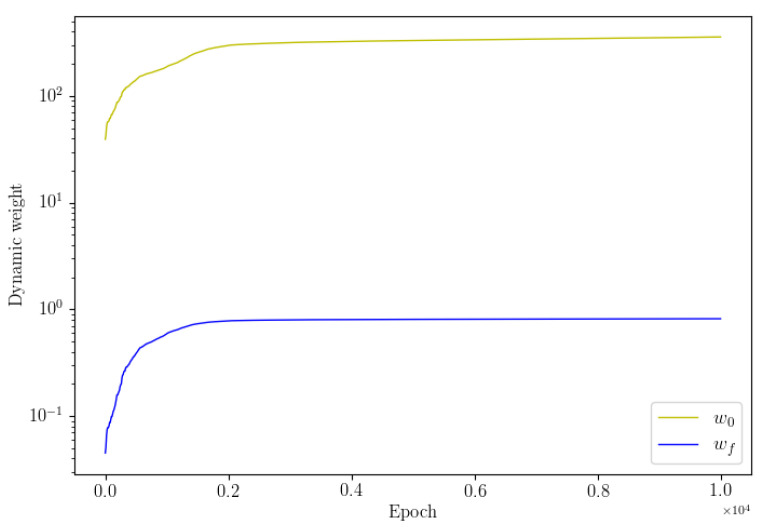
Navier–Stokes: Dynamic weights $w_{f}$, $w_{u}$ diagrams are shown.

**Figure 3 entropy-24-01254-f003:**
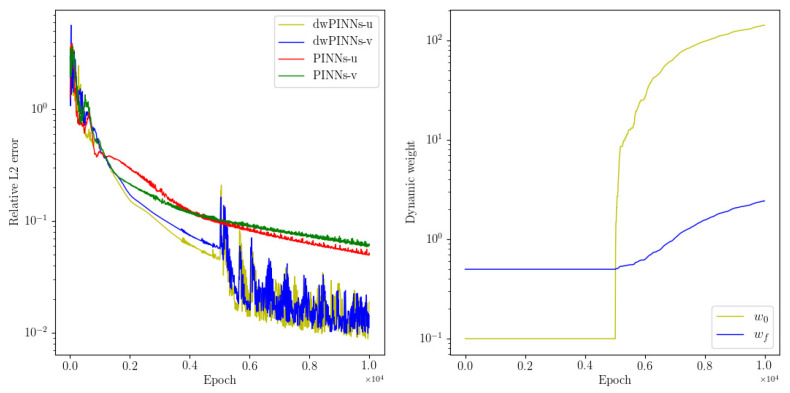
Navier–Stokes: The history of relative L2 error (**left**) of dwPINNs and PINNs and the training process of dynamic weights (**right**).

**Figure 4 entropy-24-01254-f004:**
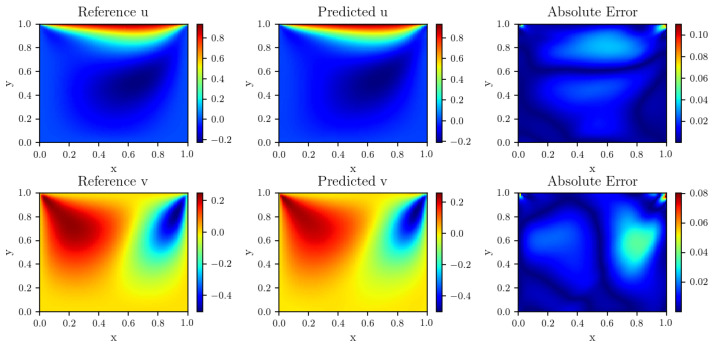
Flow in a lid-driven cavity: Reference solution using a comsol solver, prediction of dynamic weights strategy of PINNs, and absolute point-wise error. The relative L2 error of u is $6.512\times 10^{-2}$. The relative L2 error of v is $8.973\times 10^{-2}$.

**Figure 5 entropy-24-01254-f005:**
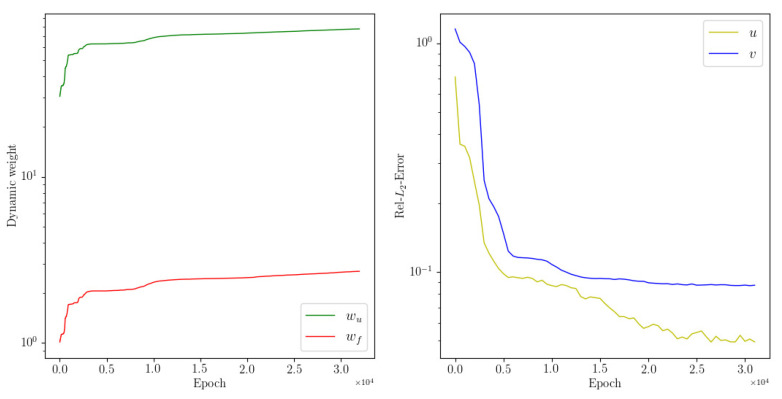
The training process of dynamic weights (**left**) and the relative L2 error of *u* and *v* (**right**).

**Figure 6 entropy-24-01254-f006:**
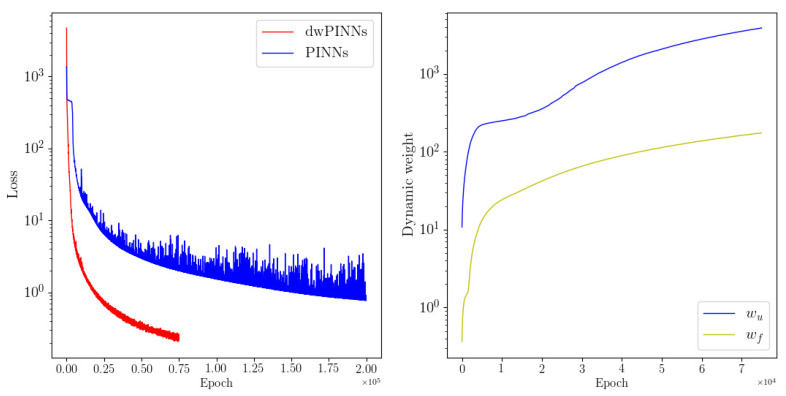
Flow past a circular cylinder: Loss history (**left**) two methods and dynamic weights history (**right**).

**Figure 7 entropy-24-01254-f007:**
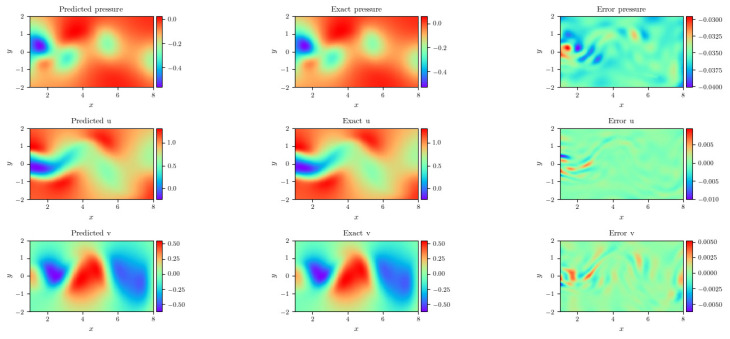
Flow past a circular cylinder: A snapshot of reference solution (**left**) with the result approximated by dwPINNs (**middle**) based on the error (**right**) at t = 10.

**Table 1 entropy-24-01254-t001:** Relative L2 Error and Training Time of dwPINNs and PINNs. The NN size is 4×50.

	Error *u*	Error *v*	Error *p*	Training Time (s)
dwPINNs	$2.890\times 10^{-3}$	$2.973\times 10^{-3}$	$5.522\times 10^{-1}$	5412.53
PINNs	$1.180\times 10^{-2}$	$1.204\times 10^{-2}$	$7.761\times 10^{-1}$	5314.67

**Table 2 entropy-24-01254-t002:** Relative L2 errors of velocity the different PINNs methods. Adam training time is 32,000. The network architecture is fixed to 4 layers with 50 neurons per hidden layer.

	dwPINNs	PINNs	SAPINNs	Learning Rate Annealing for PINNs
Relative L2 error	$6.710\times 10^{-2}$	$2.713\times 10^{-1}$	$3.415\times 10^{-1}$	$2.492\times 10^{-1}$

**Table 3 entropy-24-01254-t003:** Relative L2 error between the predicted and the exact |u| for different number of initial and boundary training data $N_{u}$, and different number of collocation points $N_{f}$. Here, the network architecture is fixed to 4 layers with 50 neurons per hidden layer.

	2000	4000	8000	10,000
200	$3.2\times 10^{-1}$	$2.7\times 10^{-1}$	$1.3\times 10^{-1}$	$1.5\times 10^{-1}$
1000	$3.1\times 10^{-1}$	$2.5\times 10^{-1}$	$9.1\times 10^{-2}$	$9.0\times 10^{-2}$
3000	$1.4\times 10^{-1}$	$1.2\times 10^{-1}$	$8.2\times 10^{-2}$	$6.7\times 10^{-2}$

**Table 4 entropy-24-01254-t004:** Relative L2 error between the predicted and the exact |u| for different numbers of hidden layers and different numbers of neurons per layer. Here, the total number of training and collocation points is fixed to $N_{u}=3000$ and $N_{f}=10,000$, respectively.

	20	30	40	50
2	$3.5\times 10^{-1}$	$1.5\times 10^{-1}$	$1.3\times 10^{-1}$	$1.5\times 10^{-1}$
3	$2.7\times 10^{-1}$	$1.3\times 10^{-1}$	$9.7\times 10^{-2}$	$8.2\times 10^{-2}$
4	$1.4\times 10^{-1}$	$1.0\times 10^{-1}$	$7.4\times 10^{-2}$	$6.7\times 10^{-2}$

**Table 5 entropy-24-01254-t005:** Relative $L_{2}$ error of velocity field *u*, *v* and relative error unknown parameters $\lambda_{1}$, $\lambda_{2}$.

	*u*	*v*	$\lambda_{1}$	$\lambda_{2}$	Training Time (s)
dwPINNs (clean)	$1.421\times 10^{-3}$	$3.832\times 10^{-3}$	0.06%	0.9%	30,574
dwPINNs (1% noise)	$2.811\times 10^{-3}$	$5.215\times 10^{-3}$	0.23%	2.1%	30,575
PINNs	$2.103\times 10^{-3}$	$6.813\times 10^{-3}$	0.99%	2.30%	51,475

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
