# Peer review of "Dynamic Weight Strategy of Physics-Informed Neural Networks for the 2D Navier–Stokes Equations"

_entropy, 2022, doi:10.3390/e24091254_

Round 1

Reviewer 1 Report

Even though the topic is really interesting and the paper have the potential to be a nice add to the literature, I found the manuscript really difficult to read. Basically, the paper needs an extensive text editing and a re-organization of the data presented. Some of the equations are not correctly defined. There are several typos, errors, etc. Foremost, the English language needs to be deeply improved.

1.     Improve use of English throughout the text.

2.     Nomenclature should be updated in the revised. Some symbols are missing and others are duplicated (including subscripts).

3.     Eqs  are hard to visualize, please present them in a clearer way. Finally, please provide an explanation for each equation and respective terms.

4.     The important contents in the abstract should be well-described, such as the purpose of the research, the principal results, and the conclusions. In addition, the authors need to explicitly describe the research objectives to make them easier for readers.

5.     INTRODUCTION:
*       The authors have presented the introduction. However, some sentences may be unclear or difficult to follow, which should be considered for reediting.
*       The authors should emphasize and explain the current study's novelty, which differs remarkably from previous research.

*       The authors need to explain that the numerical approach used in the research is one of the appropriate solutions in the context of the research problem. What are the achievements of previous studies based on a numerical basis? Also, describe what has not been achieved?

6.     MATHEMATICAL FORMULATION AND NUMERICAL SCHEME

*       It would be better if the physical model explained its parts first, followed by the appropriate computational domain. The arrows must also follow their axis proportionally.

*    If the equation comes from a particular source, it is enough to refer to it without rewriting it. Except if it is the derivation of the basic formulas.

7.     What about the validation stage? What about the quality of mesh for computation?

8.     Mathematical formulation and numerical scheme have been provided. However, it is helpful to complete the description of how to collect data, data processing scenarios, and interpret the data collection.

9.     The discussion seems inadequate, and this is too short for a reputable international journal. Many graphical presentations are similar; it is worth thinking about expressing with other graphics (if possible).

10.  CONCLUSION:
*       The conclusion must answer whether the proposed method can solve the research problem and achieve the objective. How can the numerical approach answer the existing issues?
What is the most important result? What are the implications for science and technology development?

11.  REFERENCES:
*       most of the references used are relevant to the topic and come from reliable sources, remove the reference which are not related and add some recent references.
*       The writing of some references needs to be rechecked for accuracy.

12.  What is new in this work?  Include novelty.

13.  Captions for figures and tables should be checked again. There is a significant lack of information. Please provide readers enough information on them.

14.  There are too many graphs try to reduce them or increase the discussion part.

15.  The commas and full stop in the equations should be at appropriate places.

Reviewer 2 Report

This article is devoted to the study of a dynamic weight strategy for physics-informed neural networks (dwPINNs) to balance the contribution of each loss item to the network. Three types of data for PINNs, including initial boundary value points, configuration points, observation data, the same weight is applied to similar data, while the weight is used as a penalty for loss of similar data to balance the contribution of loss terms to the parameters of the neural network. Adaptive weights are trained simultaneously with network parameters, and the data are automatically weighted in the loss function, forcing the approximation of these data to improve. This is achieved by training the network to minimize loss and maximize weights. Thus, this paper proposes a new adaptive loss balance method that dynamically adjusts the weights of loss terms during training. The validity of the method is verified by solving the forward and inverse problems of the Navier-Stokes equation.

This work undoubtedly deserves to be published in the journal of Entropy.

Round 2

Reviewer 1 Report

The writers have  have provided responses to each and every criticisms, and now  the article can be accepted in its current version.